

# Emergence of generalized hydrodynamics
# in the non-local Luttinger model

**Per Moosavi**[⋆]

Institute for Theoretical Physics, ETH Zurich,
Wolfgang-Pauli-Strasse 27, 8093 Zürich, Switzerland

⋆ pmoosavi@phys.ethz.ch

## Abstract

We propose the Luttinger model with finite-range interactions as a simple tractable example in $1+1$ dimensions to analytically study the emergence of Euler-scale hydrodynamics in a quantum many-body system. This non-local Luttinger model is an exactly solvable quantum field theory somewhere between conformal and Bethe-ansatz integrable models. Applying the recent proposal of generalized hydrodynamics, we show that the model allows for fully explicit yet non-trivial solutions of the resulting Euler-scale hydrodynamic equations. Comparing with exact analytical non-equilibrium results valid at all time and length scales, we show perfect agreement at the Euler scale when the interactions are short range. A formal proof of the emergence of generalized hydrodynamics in the non-local Luttinger model is also given, and effects of long-range interactions are briefly discussed.



# 1    Introduction

Recent years have witnessed new advances in the application of hydrodynamics to study 1+1-dimensional quantum (and classical) many-body systems out of equilibrium. One such development is *generalized hydrodynamics* (GHD) [1,2], which extends conventional hydrodynamics by taking into account the macroscopically large number of conserved charges characteristic of integrable systems, see [3] for a pedagogical overview. Since proposed, GHD has attracted considerable interest, see, e.g., [4–20], and its predictions were recently verified experimentally [21]. However, so far, except for the classical hard-rod gas in [22], there are no other rigorous proofs of emergence of hydrodynamics.

The idea behind GHD is the same as for conventional hydrodynamics: One studies expectations of observables with respect to local-equilibrium states within fluids cells, which are assumed to describe the exact expectations at appropriate time and length scales. Simply put, the key assumption is a reduction of degrees of freedom with increasing scales. The difference between generalized and conventional hydrodynamics is the choice of statistical ensemble: The former uses a generalized Gibbs ensemble, see, e.g., [23], while the latter uses an ordinary one. The coarsest hydrodynamic description is that on the Euler scale, which neglects higher derivatives and thus dissipative terms or other corrections; we will exclusively consider this scale in the present paper.

Most GHD examples, including those in [1,2], involve the Bethe ansatz, which has the advantage of applicability to a large family of models, i.e., those that are Bethe-ansatz integrable. However, one disadvantage is that the results are not explicit as one is left to numerically solve certain integral equations. For quantum systems, the exception in the literature is conformal field theory (CFT) in 1+1 dimensions, but pure CFT is instead too simple from the point of view of Euler-scale GHD while deformed ones require approximate solutions, cf. [24].

Fortunately, there is a 1+1-dimensional quantum many-body system somewhere between CFTs and Bethe-ansatz integrable models, namely the Luttinger model [25–27] with finite-range interactions breaking conformal invariance [28–32]. This *non-local Luttinger* (NLL) model describes interacting massless fermions and is exactly solvable by bosonization. To formally state the model on the circle with length $L$, let $\psi_r^-(x)$ and $\psi_r^+(x) = \psi_r^-(x)^\dagger$ for $x \in [-L/2, L/2]$ and $r = +(-)$ denote right- (left-) moving fermionic fields satisfying canonical anti-commutation relations

$$\{\psi_r^-(x), \psi_{r'}^+(x')\} = \delta_{r,r'}\delta(x - x'), \quad \{\psi_r^\pm(x), \psi_{r'}^\pm(x')\} = 0 \tag{1}$$

and anti-periodic boundary conditions $\psi_r^\pm(-L/2) = -\psi_r^\pm(L/2)$. The NLL Hamiltonian is

$$
\begin{aligned}
H = &\sum_{r=\pm} \int_{-L/2}^{L/2} dx \,{:}\psi_r^+(x)\left(-\mathrm{i}r v_F \partial_x\right)\psi_r^-(x){:} \\
&+ \frac{1}{2} \sum_{r,r'=\pm} \int_{-L/2}^{L/2} dx \int_{-L/2}^{L/2} dx' \left[\delta_{r,-r'} g_2 V_2(x-x') + \delta_{r,r'} g_4 V_4(x-x')\right]\rho_r(x)\rho_{r'}(x'),
\end{aligned} \tag{2}
$$

where $v_F > 0$ is the Fermi velocity, $\rho_\pm(x) = {:}\psi_\pm^+(x)\psi_\pm^-(x){:}$ denote fermion densities, $g_{2,4}$ are dimensionless coupling constants, and $V_{2,4}(x)$ are finite-range interaction potentials. Here, ${:}\cdots{:}$ indicates Wick ordering with respect to the vacuum $|\Psi_0\rangle$ that is the ground state of the non-interacting Hamiltonian $H\big|_{g_2=g_4=0}$. The potentials are required to satisfy certain conditions for the model to be well-defined (see Sec. 2). We will restrict ourselves to short-range interactions. Thus, there is a finite length scale, the interaction range, which by definition is absent in the conformal case of point-like interactions. For later reference, we mention that long-range interactions also satisfy the conditions and thus are possible in principle.

The NLL model was previously studied out of equilibrium in [31, 32] by exact analytical means and shown to exhibit dispersive effects that are not present in the conformal case. To see why, we recall that bosonization allows one to map the NLL model to a quasi-free model of bosons and there identify the excitations, commonly called plasmons. The reason for the dispersive effects is that the plasmon modes, while decoupled (as usual for a quasi-free model), propagate with a momentum-dependent velocity that depends non-trivially on the strength and range of the NLL interactions. This gives rise to additional structure compared to CFT, and thus it is natural to expect a richer hydrodynamic description. (In the limit of point-like interactions, conformal invariance is recovered, manifested by that the propagation velocity is the same for all plasmon modes.)

In this paper, we present a fully explicit application of GHD to the NLL model with short-range interactions. This model is proposed as a simple tractable example to investigate the emergence of Euler-scale GHD in a quantum many-body system. To illustrate this, we will consider the time evolution of local operators $\mathcal{O}(t) = e^{iHt}\mathcal{O}e^{-iHt}$ under $H$ in (2) starting from non-equilibrium initial states defined by smooth $L$-periodic inverse-temperature and chemical-potential profiles $\beta(x) > 0$ and $\mu(x)$, respectively. (The latter generalize the usual constant thermodynamic variables that correspond to equilibrium states.) This is an example of an inhomogeneous quantum quench. Such non-equilibrium results in the NLL model can be computed by exact analytical means [31, 32], which allows for direct analytical comparisons between those and the GHD results that we will derive here.

The case with both $\beta(x)$ and $\mu(x)$ is quite involved, but key aspects are captured already when the former is constant. Thus, for simplicity, let $\beta(x) = \beta > 0$ for now and consider the initial state $\hat{\rho}_{\mu(\cdot)} = Z_{\mu(\cdot)}^{-1}\exp\big(-\beta\big[H - \int_{-L/2}^{L/2}dx\,\mu(x)\rho(x)\big]\big)$ with a smooth $\mu(x)$, where $\rho(x) = \rho_+(x) + \rho_-(x)$ is the total particle density and $Z_{\mu(\cdot)}^{-1}$ is the normalization. Associated to $\rho(x)$ is the charge current $j(x)$ satisfying $\partial_t\rho + \partial_x j = 0$; they are the components of a conserved U(1) current present in the model. The exact analytical results for the expectations of these operators in the thermodynamic limit $L \to \infty$ are [31]

$$\lim_{L\to\infty}\mathrm{Tr}\big[\hat{\rho}_{\mu(\cdot)}\rho(x,t)\big] = \int_{-\infty}^{\infty}\frac{dp}{2\pi}\frac{K(p)\mu(p)}{2\pi v(p)}\big[e^{ip[x-v(p)t]} + e^{ip[x+v(p)t]}\big], \tag{3a}$$

$$\lim_{L\to\infty}\mathrm{Tr}\big[\hat{\rho}_{\mu(\cdot)}j(x,t)\big] = \int_{-\infty}^{\infty}\frac{dp}{2\pi}\frac{K(p)\mu(p)}{2\pi}\big[e^{ip[x-v(p)t]} - e^{ip[x+v(p)t]}\big], \tag{3b}$$

where $v(p)$ and $K(p)$ denote the propagation velocity and the Luttinger parameter, respectively, and $\mu(p)$ is the Fourier transform of $\mu(x)$. We will see that $v(p)$ and $K(p)$ depend non-trivially on momentum $p$ through their dependence on the Fourier transforms of $g_{2,4}V_{2,4}(x)$ in (2) [see (7)]. The corresponding Euler-scale GHD results describe expectations within fluid cells that are in local equilibrium. Such states $\hat{\rho}_{\boldsymbol{\beta}(x,t)}$ are given by a suitably chosen generalized Gibbs ensemble consisting of conserved charges $\boldsymbol{Q} = (Q_1, Q_2, \ldots)$, which are assumed to be (sufficiently) local, with conjugate thermodynamic fields $\boldsymbol{\beta}(x,t) = (\beta_1(x,t), \beta_2(x,t), \ldots)$ that depend on the spacetime point $(x,t)$ of the fluid cell. Given the initial state $\hat{\rho}_{\mu(\cdot)}$ above, it will follow from our general results that the only non-constant fields in $\boldsymbol{\beta}(x,t)$ are $\mu_\pm(x,t) = \mu(x \mp v(0)t)$ conjugate to the conserved U(1) charges for right- and left-moving

excitations. For this case, we will show that the GHD results for the particle density and the charge current are

$$\lim_{L\to\infty} \text{Tr}\big[\hat{\rho}_{\boldsymbol{\beta}(x,t)}\rho(0,0)\big] = \frac{K(0)\big[\mu(x-v(0)t)+\mu(x+v(0)t)\big]}{2\pi v(0)}, \qquad (4a)$$

$$\lim_{L\to\infty} \text{Tr}\big[\hat{\rho}_{\boldsymbol{\beta}(x,t)}j(0,0)\big] = \frac{K(0)\big[\mu(x-v(0)t)-\mu(x+v(0)t)\big]}{2\pi}. \qquad (4b)$$

Clearly, (3) and (4) are not the same. For example, (3) allow for non-rigid propagation of wave packets (dispersion) while only rigid propagation is possible in (4). However, one can show that the GHD results emerge from the former at the Euler scale if the interactions are short range. More precisely, $\lim_{\lambda\to\infty}\lim_{L\to\infty}\text{Tr}\big[\hat{\rho}_{\mu(\cdot/\lambda)}\rho(\lambda x,\lambda t)\big] = \lim_{L\to\infty}\text{Tr}\big[\hat{\rho}_{\boldsymbol{\beta}(x,t)}\rho(0,0)\big]$ with $\mu_{\pm}(x,t) = \mu(x\mp v(0)t)$, and similarly for $j(x,t)$. This exemplifies the emergence of Euler-scale GHD in the NLL model, while such a description would always be true for charge transport in the case of point-like interactions [since then $v(p) = v(0)$ and $K(p) = K(0)$ for all $p$].[1]

The rest of the paper is organized as follows. In Sec. 2, the relevant conserved charges for the NLL model are identified based on its exact solution by bosonization. These will define our generalized Gibbs ensemble. In Sec. 3, following [3], the associated hydrodynamic equations are derived and solved exactly. In Sec. 4, these solutions are used to compute fully explicit GHD results for heat and charge transport, which then are compared with the corresponding exact analytical ones in [31, 32] valid at all time and length scales. As discussed above, we show that the two results are in perfect agreement at the Euler scale, confirming the emergence of GHD, at least as far as heat and charge transport is concerned. A general but formal proof of the emergence of Euler-scale GHD in the NLL model is given in Sec. 5. Concluding remarks are given in Sec. 6, including a brief discussion of effects of long-range interactions.

## 2 Bosonization and generalized Gibbs ensemble

The presentation below follows [31] (using the conventions in [28, 29] for the interactions) and briefly summarizes the solution of the NLL model by bosonization for the purpose of identifying the relevant conserved charges to be included in our generalized Gibbs ensemble.

### 2.1 Exact solution by bosonization

For the NLL model, it is more practical to work in momentum space. It is also in this way that we would make the model mathematically precise, see [31, 33], but here we will not go further into such matters. To this end, let $V_{2,4}(p) = \int_{-L/2}^{L/2} dx\, V_{2,4}(x)e^{-ipx}$ and $\rho_{\pm}(p) = \int_{-L/2}^{L/2} dx\, \rho_{\pm}(x)e^{-ipx}$ for momenta $p \in (2\pi/L)\mathbb{Z}$.[2] The conditions on the NLL interactions can then be expressed as

$$V_{2,4}(p) = V_{2,4}(-p), \quad \big|g_2 V_2(p)\big| < 2\pi v_F + g_4 V_4(p) \quad \forall p, \quad \sum_{p>0} \frac{p\big[g_2 V_2(p)\big]^2}{2\pi v_F\big[2\pi v_F + g_4 V_4(p)\big]} < \infty. \qquad (5)$$

Examples of possible potentials include $V_{2,4}(p) = \pi v_F/[1+(ap)^2]$ and $V_{2,4}(p) = \pi v_F \,\text{sech}(ap)$ with interaction range $a > 0$.

---

[1]For point-like interactions, there is already a difference between the exact analytical results for heat transport and the GHD ones due to the presence of a Schwarzian-derivative term, see [32], which vanishes at the Euler scale.

[2]Note that the latter Fourier transform is formal.

The upshot of bosonization is that the NLL Hamiltonian can be written as a bilinear in the densities $\rho_\pm(p)$. The latter can be shown to satisfy

$$\rho_\pm(p)^\dagger = \rho_\pm(-p), \quad \rho_+(p)|\Psi_0\rangle = \rho_-(-p)|\Psi_0\rangle = 0 \quad \forall p \geq 0,$$
$$\left[\rho_r(p), \rho_{r'}(-p')\right] = r\delta_{r,r'}\frac{Lp}{2\pi}\delta_{p,p'}, \tag{6}$$

which also defines the vacuum $|\Psi_0\rangle$. For details on the construction of the Hilbert space, see, e.g., [33]. Since the operators $\rho_\pm(p)$ satisfy (non-trivial) commutation relations, they are bosonic.

To be more explicit, in momentum space, the bosonized version of the formal Hamiltonian in (2) can be written using the renormalized Fermi velocity and the Luttinger parameter

$$v(p) = v_F\sqrt{\left[1 + \frac{g_4 V_4(p)}{2\pi v_F}\right]^2 - \left[\frac{g_2 V_2(p)}{2\pi v_F}\right]^2}, \quad K(p) = \sqrt{\frac{2\pi v_F + g_4 V_4(p) - g_2 V_2(p)}{2\pi v_F + g_4 V_4(p) + g_2 V_2(p)}}, \tag{7}$$

which depend on momentum if the potentials are finite range. Indeed, one can show that

$$H = \sum_{r,r'}\sum_p \frac{\pi}{L}v(p)\frac{1 + rr'K(p)^2}{2K(p)}:\rho_r(-p)\rho_{r'}(p): - \sum_{p>0}\left[v_F - v(p)\frac{1 + K(p)^2}{2K(p)}\right]p. \tag{8}$$

This Hamiltonian can be written in diagonal form using

$$\tilde{\rho}_r(p) = \sum_{r'}\frac{1 + rr'K(p)}{2\sqrt{K(p)}}\rho_{r'}(p). \tag{9}$$

For $p \neq 0$, these are obtained by a Bogoliubov transformation $\tilde{\rho}_r(p) = e^{-iS}\rho_r(p)e^{iS}$ implemented by a unitary operator $e^{iS}$, cf., e.g., [31] for details. For $p = 0$, let $\tilde{Q}_r = \tilde{\rho}_r(p = 0)$. We recall that $\tilde{\rho}_r(p)$ are commonly referred to as plasmon operators. The result is

$$H = \sum_r \frac{\pi}{L}v(0)\tilde{Q}_r^2 + \sum_r\sum_{p\neq 0}\frac{\pi}{L}v(p):\tilde{\rho}_r(-p)\tilde{\rho}_r(p): + E_{\text{GS}}, \tag{10}$$

where $E_{\text{GS}} = \sum_{p>0}[v(p) - v_F]p$ is the energy of the ground state $|\Psi\rangle = e^{-iS}|\Psi_0\rangle$ of $H$ and, by abuse of notation, $:\cdots:$ indicates Wick ordering with respect to $|\Psi\rangle$. We will continue to abuse notation in this way in what follows.

To see that the Hamiltonian in (10) is diagonal, we introduce boson creation and annihilation operators $\tilde{b}_p^+ = (\tilde{b}_p^-)^\dagger$ and $\tilde{b}_p^- = -i\sqrt{2\pi/L|p|}\tilde{\rho}_+(p)$ if $p > 0$ while $\tilde{b}_p^- = i\sqrt{2\pi/L|p|}\tilde{\rho}_-(p)$ if $p < 0$. This allows us to write $H = \sum_r \pi v(0)\tilde{Q}_r^2/L + \sum_r\sum_{p>0}\omega(p)\tilde{b}_{rp}^+\tilde{b}_{rp}^- + E_{\text{GS}}$ with the dispersion relation

$$\omega(p) = v(p)p \tag{11}$$

obtained from $v(p)$ in (7).

## 2.2 Conserved charges

Let $\mathcal{I}_1 = \{(r,p)\,|\,r \in \{+,-\},\, p \in (2\pi/L)\mathbb{Z}^+\}$ and $\mathcal{I}_0 = \{(r,0)\,|\,r \in \{+,-\}\}$ as well as $\mathcal{I} = \mathcal{I}_0 \cup \mathcal{I}_1$. For the non-zero modes, the conserved charges and their corresponding densities can be written[3]

$$Q_{r',p'} = q_{r',p'}(p = 0),$$
$$q_{r',p'}(p) = \frac{\pi}{L}v(p')\left[:\tilde{\rho}_{r'}(p-p')\tilde{\rho}_{r'}(p'): + :\tilde{\rho}_{r'}(-p')\tilde{\rho}_{r'}(p+p'):\right] \quad \forall(r',p') \in \mathcal{I}_1. \tag{12}$$

---

[3]The latter are symmetrized to make manifest that $q_{r',p'}(p)^\dagger = q_{r',p'}(-p)$.

For the zero modes, using $\tilde{\rho}_\pm(p)$ in (9), we identify

$$Q^J_{r'} = q^J_{r'}(p=0), \quad q^J_{r'}(p) = \sqrt{K(p)}\tilde{\rho}_{r'}(p) \quad \forall r' = \pm \tag{13}$$

as the charges and densities associated to the conserved $U(1)$ current.[4] In addition to these, we also need to introduce[5]

$$Q_{r',0} = q_{r',0}(p=0), \quad q_{r',0}(p) = \frac{\pi}{L}v(0)\tilde{\rho}_{r'}(p)\tilde{Q}_{r'} \quad \forall r' = \pm, \tag{14}$$

where we recall that $\tilde{Q}_\pm = \tilde{\rho}_\pm(p=0)$. The $Q_{r',0}$ account for the energy contribution from the relevant charge sector of the Hilbert space, but they will not contribute to our results in the thermodynamic limit, see Appendix A.

Define $\mathbf{Q} = ((Q_{r',p'})_{(r',p')\in\mathcal{I}}, (Q^J_{r'})_{r'=\pm})$. That this set forms a family of mutually commuting conserved charges follows from that $H = \sum_{(r',p')\in\mathcal{I}} Q_{r',p'} + E_{GS}$ together with (6) and (9). The set $\mathbf{Q}$ will define our generalized Gibbs ensemble.[6]

## 2.3 Densities and currents

To study the dynamics under $H$ in (10), define $\tilde{\rho}_\pm(p,t) = e^{iHt}\tilde{\rho}_\pm(p)e^{-iHt}$. It follows from (6) and (9) that $\tilde{\rho}_\pm(p,t) = \tilde{\rho}_\pm(p)e^{\mp i\omega(p)t}$ with $\omega(p)$ in (11).

For each $Q_{r',p'}$ in $\mathbf{Q}$, let $q_{r',p'}(p,t)$ and $j_{r',p'}(p,t)$ denote the corresponding time-dependent densities and currents. In momentum space, these must satisfy $\partial_t q_{r',p'}(p,t) + ip j_{r',p'}(p,t) = 0$. It follows that[7]

$$q_{r',p'}(p,t) = \frac{\pi}{L}v(p')\big[:\tilde{\rho}_{r'}(p-p',t)\tilde{\rho}_{r'}(p',t): + :\tilde{\rho}_{r'}(-p',t)\tilde{\rho}_{r'}(p+p',t):\big], \tag{15a}$$

$$j_{r',p'}(p,t) = \frac{\pi}{L}v(p')r'\big[v(p-p',p'):\tilde{\rho}_{r'}(p-p',t)\tilde{\rho}_{r'}(p',t):$$
$$+ v(-p',p+p'):\tilde{\rho}_{r'}(-p',t)\tilde{\rho}_{r'}(p+p',t):\big] \tag{15b}$$

for $(r',p') \in \mathcal{I}_1$ and

$$q_{r',0}(p,t) = \frac{\pi}{L}v(0)\tilde{\rho}_{r'}(p,t)\tilde{Q}_{r'}, \tag{15c}$$

$$j_{r',0}(p,t) = \frac{\pi}{L}v(0)r'v(p,0)\tilde{\rho}_{r'}(p,t)\tilde{Q}_{r'} \tag{15d}$$

for $r' = \pm$, where

$$v(p_1,p_2) = \frac{\omega(p_1) + \omega(p_2)}{p_1 + p_2} \quad (p_1+p_2 \neq 0), \qquad v(-p,p) = v^g(p) \tag{16}$$

with

$$v^g(p) = d\omega(p)/dp. \tag{17}$$

The interpretation of $v^g(p)$ is as the group velocity corresponding to $\omega(p)$ in (11).

For the two $Q^J_{r'}$ in $\mathbf{Q}$, the corresponding time-dependent densities and currents are

$$q^J_{r'}(p,t) = \sqrt{K(p)}\tilde{\rho}_{r'}(p,t), \qquad j^J_{r'}(p,t) = r'v(p)q^J_{r'}(p,t), \tag{18}$$

respectively, which satisfy $\partial_t q^J_{r'}(p,t) + ip j^J_{r'}(p,t) = 0$.

---

[4]Using notation common in CFT, $J^\pm_n = q^J_\pm(\pm 2\pi n/L)$ for $n \in \mathbb{Z}$ are the generators of a "generalized" double $\mathfrak{u}(1)$ current algebra satisfying $[J^\pm_n, J^\pm_m] = \kappa_n n\delta_{n+m,0}$ and $[J^\pm_n, J^\mp_m] = 0$ with $\kappa_n = K(2\pi n/L)$.

[5]One reason is that $Q_{\pm,0}$ appear in the Hamiltonian, see (10). Another is that they are needed for making certain expectations well defined as the zero modes otherwise could yield divergent contributions, cf. (47a) in Appendix A.

[6]While the $Q_{r',p'}$ charges do not appear local in the bosonic picture, they are mutually commuting terms in the NLL Hamiltonian $H$ which should be sufficiently local in the fermionic picture for short-range interactions.

[7]It is manifest that $q_{r',p'}(p,t)^\dagger = q_{r',p'}(-p,t)$ and $j_{r',p'}(p,t)^\dagger = j_{r',p'}(-p,t)$.

# 3 Generalized hydrodynamics

Let $\boldsymbol{\beta} = ((\beta_{r',p'})_{(r',p')\in\mathcal{I}},(\mu_{r'}^{J})_{r'=\pm})$ with $\beta_{r',p'} > 0$ and $\mu_{r'}^{J} \in \mathbb{R}$ denote the thermodynamic variables conjugate to the charges in $\boldsymbol{Q}$ introduced in Sec. 2.2, and let $\boldsymbol{\beta}(x)$ denote the corresponding set of $L$-periodic thermodynamic fields $\beta_{r',p'}(x) > 0$ and $\mu_{r'}^{J}(x) \in \mathbb{R}$. (The latter are smooth position-dependent profiles generalizing the thermodynamic variables.) Define

$$G_{\boldsymbol{\beta}(\cdot)} = \sum_{(r',p')\in\mathcal{I}_1} \int_{-L/2}^{L/2} \mathrm{d}x\, \beta_{r',p'}(x)q_{r',p'}(x) + \sum_{r'=\pm} \int_{-L/2}^{L/2} \mathrm{d}x\, \beta_{r',0}(x)\big[q_{r',0}(x) - \mu_{r'}^{J}(x)q_{r'}^{J}(x)\big],$$

(19)

where $q_{r',p'}(x) = \sum_p L^{-1}q_{r',p'}(p)\mathrm{e}^{\mathrm{i}px}$ and similarly for $q_{r'}^{J}(x)$, and the expectation

$$\langle\cdots\rangle_{\boldsymbol{\beta}(\cdot)} = \frac{\mathrm{Tr}\big[\mathrm{e}^{-G_{\boldsymbol{\beta}(\cdot)}}(\cdots)\big]}{\mathrm{Tr}\big[\mathrm{e}^{-G_{\boldsymbol{\beta}(\cdot)}}\big]}.$$

(20)

This defines an inhomogeneous initial state of the form in [31,32,34]. As explained in Sec. 1, we are interested in this expectation of time-evolved local observables $\mathcal{O}(x,t) = \mathrm{e}^{\mathrm{i}Ht}\mathcal{O}(x)\mathrm{e}^{-\mathrm{i}Ht}$ under the dynamics given by $H$ in (10), i.e., we are interested in $\langle\mathcal{O}(x,t)\rangle_{\boldsymbol{\beta}(\cdot)}$.

## 3.1 Euler-scale GHD

Given the initial state defined by (19) with smooth profiles $\boldsymbol{\beta}(\cdot)$, we consider the Euler-scale hydrodynamic approximation [3,35]

$$\langle\mathcal{O}(x,t)\rangle_{\boldsymbol{\beta}(\cdot)} \approx \langle\mathcal{O}\rangle_{\boldsymbol{\beta}(x,t)}$$

(21)

of the expectation in (20) for any local observable $\mathcal{O}(x,t) = \mathrm{e}^{\mathrm{i}Ht}\mathcal{O}(x)\mathrm{e}^{-\mathrm{i}Ht}$. On the r.h.s., we have introduced

$$\langle\mathcal{O}\rangle_{\boldsymbol{\beta}(x,t)} = \frac{\mathrm{Tr}\big[\mathrm{e}^{-G_{\boldsymbol{\beta}(x,t)}}\mathcal{O}\big]}{\mathrm{Tr}\big[\mathrm{e}^{-G_{\boldsymbol{\beta}(x,t)}}\big]}$$

(22)

for $\mathcal{O} = \mathcal{O}(0,0)$ with

$$G_{\boldsymbol{\beta}(x,t)} = \sum_{(r',p')\in\mathcal{I}_1} \beta_{r',p'}(x,t)Q_{r',p'} + \sum_{r'=\pm} \beta_{r',0}(x,t)\big[Q_{r',0} - \mu_{r'}^{J}(x,t)Q_{r'}^{J}\big]$$

(23)

and time-dependent thermodynamic fields $\boldsymbol{\beta}(x,t) = ((\beta_{r',p'}(x,t))_{(r',p')\in\mathcal{I}},(\mu_{r'}^{J}(x,t))_{r'=\pm})$ conjugate to the charges in $\boldsymbol{Q}$ and satisfying $\boldsymbol{\beta}(x,0) = \boldsymbol{\beta}(x)$. In words, these fields describe the local-equilibrium state within a fluid cell at spacetime point $(x,t)$.[8] As before, we require that $\beta_{r',p'}(x,t) > 0$ and $\mu_{r'}^{J}(x,t) \in \mathbb{R}$.

## 3.2 Hydrodynamic equations

Given our generalized Gibbs ensemble $\boldsymbol{Q}$, following [3], the associated Euler-scale hydrodynamic equations are

$$\partial_t\langle q_{r',p'}\rangle_{\boldsymbol{\beta}(x,t)} + \sum_{(r'',p'')\in\mathcal{I}} A_{r',p'}^{r'',p''}(x,t)\partial_x\langle q_{r'',p''}\rangle_{\boldsymbol{\beta}(x,t)} = 0, \quad A_{r',p'}^{r'',p''}(x,t) = \frac{\partial\langle j_{r',p'}\rangle_{\boldsymbol{\beta}(x,t)}}{\partial\langle q_{r'',p''}\rangle_{\boldsymbol{\beta}(x,t)}}$$

(24a)

---

[8]We note that, in principle, $\mathcal{O}(x,t)$ could depend on additional spacetime points. In other words, $\mathcal{O}(x,t) = \mathcal{O}(x,t;x_1,t_1;\ldots;x_n,t_n)$, in which case $\mathcal{O} = \mathcal{O}(0,0;\Delta x_1,\Delta t_1;\ldots;\Delta x_n,\Delta t_n)$ with $\Delta x_j = x_j - x$ and $\Delta t_j = t_j - t$. For clarity, we stress that the latter should be assumed fixed, i.e., $\Delta x_j$ and $\Delta t_j$ should not scale with $\lambda$ in (37).

for $(r',p'),(r'',p'') \in \mathcal{I}$ and

$$\partial_t \langle q_{r'}^J \rangle_{\boldsymbol{\beta}(x,t)} + \sum_{r''=\pm} A_{r'}^{r''}(x,t)\partial_x \langle q_{r''}^J \rangle_{\boldsymbol{\beta}(x,t)} = 0, \quad A_{r'}^{r''}(x,t) = \frac{\partial \langle j_{r'}^J \rangle_{\boldsymbol{\beta}(x,t)}}{\partial \langle q_{r''}^J \rangle_{\boldsymbol{\beta}(x,t)}} \tag{24b}$$

for $r',r'' = \pm$, where $A_{r',p'}^{r'',p''}(x,t)$ and $A_{r'}^{r''}(x,t)$ are referred to as flux Jacobians. To solve these equations, we note that (15) implies

$$\langle q_{r',p'} \rangle_{\boldsymbol{\beta}(x,t)} = \sum_p \frac{1}{L} \langle q_{r',p'}(p) \rangle_{\boldsymbol{\beta}(x,t)} = \frac{1}{L} \langle Q_{r',p'} \rangle_{\boldsymbol{\beta}(x,t)}, \tag{25a}$$

$$\langle j_{r',p'} \rangle_{\boldsymbol{\beta}(x,t)} = r'v^g(p') \langle q_{r',p'} \rangle_{\boldsymbol{\beta}(x,t)} \tag{25b}$$

with the group velocity in (17), while (18) implies

$$\langle q_{r'}^J \rangle_{\boldsymbol{\beta}(x,t)} = \sum_p \frac{1}{L} \langle q_{r'}^J(p) \rangle_{\boldsymbol{\beta}(x,t)} = \frac{1}{L} \langle Q_{r'}^J \rangle_{\boldsymbol{\beta}(x,t)}, \tag{26a}$$

$$\langle j_{r'}^J \rangle_{\boldsymbol{\beta}(x,t)} = r'v(0) \langle q_{r'}^J \rangle_{\boldsymbol{\beta}(x,t)}. \tag{26b}$$

(In the equations above, we used that only the $p = 0$ contribution survives.) It follows that

$$A_{r',p'}^{r'',p''}(x,t) = r'v^g(p')\delta_{r',r''}\delta_{p',p''} = v_{r',p'}^{\text{eff}}\delta_{r',r''}\delta_{p',p''}, \tag{27a}$$

$$A_{r'}^{r''}(x,t) = r'v(0)\delta_{r',r''} = v_{r',0}^{\text{eff}}\delta_{r',r''} \tag{27b}$$

with the effective velocity

$$v_{r',p'}^{\text{eff}} = r'v^g(p'). \tag{28}$$

We note that (17) implies $v^g(0) = v(0)$ and that $v_{r',p'}^{\text{eff}}$ depends non-trivially on the momentum $p' \geq 0$ via (17) and (7), i.e., in general, it is different for each plasmon mode with sign depending on $r'$. It is manifest that $A_{r',p'}^{r'',p''}(x,t) = A_{r',p'}^{r'',p''}$ and $A_{r'}^{r''}(x,t) = A_{r'}^{r''}$ are diagonal[9] and independent of $(x,t)$, and thus obviously independent of $\langle q_{r',p'} \rangle_{\boldsymbol{\beta}(x,t)}$ and $\langle q_{r'}^J \rangle_{\boldsymbol{\beta}(x,t)}$, respectively. This simplifies the treatment considerably compared to the general situation, see, e.g., [3].

From (24) and (27), the Euler-scale hydrodynamic equations for the NLL model become

$$\partial_t \langle q_{r',p'} \rangle_{\boldsymbol{\beta}(x,t)} + v_{r',p'}^{\text{eff}}\partial_x \langle q_{r',p'} \rangle_{\boldsymbol{\beta}(x,t)} = 0, \quad \partial_t \langle q_{r'}^J \rangle_{\boldsymbol{\beta}(x,t)} + v_{r',0}^{\text{eff}}\partial_x \langle q_{r'}^J \rangle_{\boldsymbol{\beta}(x,t)} = 0 \tag{29}$$

with $v_{r',p'}^{\text{eff}}$ in (28), where we recall that $q_{r',p'} = q_{r',p'}(x = 0, t = 0)$ and similarly for $q_{r'}^J$. We stress that (29) are differential equations for $\boldsymbol{\beta}(x,t)$ with initial conditions $\boldsymbol{\beta}(x,0) = \boldsymbol{\beta}(x) = ((\beta_{r',p'}(x))_{(r',p')\in\mathcal{I}}, (\mu_{r'}^J(x))_{r'=\pm})$. In Appendix A, we show that the solutions are

$$\beta_{r',p'}(x,t) = \beta_{r',p'}(x - v_{r',p'}^{\text{eff}}t), \quad \mu_{r'}^J(x,t) = \mu_{r'}^J(x - v_{r',0}^{\text{eff}}t). \tag{30}$$

These give the fully explicit spacetime dependence of the thermodynamic fields in (23) that define the local-equilibrium state within a fluid cell.

We note that all conserved charges in $\boldsymbol{Q}$ are always involved, unless if $\beta_{r',p'}(x) = \beta_{r',p'} \to \infty$ for $(r',p') \in \mathcal{I}_1$, in which case the corresponding mode with charge $Q_{r',p'}$ is in its ground state, or if $\mu_{r'}(x) = 0$, in which case the corresponding $Q_{r'}^J$ does not play a role.

It follows from (23) together with (30) that modes can propagate with their own velocity $v_{r',p'}^{\text{eff}}$. As in [31,32], this implies that dispersive effects are possible in the NLL model. This is not the case for point-like interactions, where transport is purely ballistic and wave packets

---

[9]Our densities $q_{r',p'}(x)$ and $q_{r'}^J(x)$ correspond to normal modes, which in the general case are obtained by first diagonalizing the flux Jacobians.

propagate rigidly. Here, transport is still purely ballistic (since individual plasmon modes propagate rigidly and are decoupled from each other). However, we will see that wave packets for heat transport propagate non-rigidly in general, leading to dispersion, while for charge transport the GHD results do not exhibit dispersion, different from the exact analytical results in [31].

# 4 Heat and charge transport

We begin by stating the density and current operators for heat and charge transport in the NLL model, see [31, 32]. In what follows, recall that $\tilde{\rho}_{\pm}(p,t) = \tilde{\rho}_{\pm}(p)e^{\mp i\omega(p)t}$ with $\omega(p)$ in (11).

The energy density and heat current operators are

$$\mathcal{E}(x,t) = \sum_{r,r'}\sum_{p,p'}\frac{\pi}{L^2}v_{r,r'}(p-p',p')\colon\!\tilde{\rho}_r(p-p',t)\tilde{\rho}_{r'}(p',t)\!\colon e^{ipx} + \mathcal{E}_{\text{GS}}, \tag{31a}$$

$$\mathcal{J}(x,t) = \sum_{r,r'}\sum_{p,p'}\frac{\pi}{L^2}u_{r,r'}(p-p',p')v_{r,r'}(p-p',p')\colon\!\tilde{\rho}_r(p-p',t)\tilde{\rho}_{r'}(p',t)\!\colon e^{ipx}, \tag{31b}$$

where the sums over $p$ and $p'$ range over $(2\pi/L)\mathbb{Z}$, $\mathcal{E}_{\text{GS}} = E_{\text{GS}}/L$ is the ground-state energy density [cf. (10)], and

$$v_{r_1,r_2}(p_1,p_2) = \frac{r_1 v(p_1) + r_2 v(p_2)}{2}\frac{r_1 K(p_2) + r_2 K(p_1)}{2\sqrt{K(p_1)K(p_2)}}, \tag{32a}$$

$$u_{r_1,r_2}(p_1,p_2) = \frac{r_1\omega(p_1) + r_2\omega(p_2)}{p_1 + p_2} \quad (p_1 + p_2 \neq 0), \qquad u_{r,r}(-p,p) = rv^g(p) \tag{32b}$$

with $v^g(p)$ in (17). Note that $v_{r_1,r_2}(-p,p) = v_{r_1,r_2}(p,p) = \delta_{r_1,r_2}v(p)$.

The particle density and charge current operators are

$$\rho(x,t) = \sum_p\frac{1}{L}\sqrt{K(p)}\big[\tilde{\rho}_+(p,t) + \tilde{\rho}_-(p,t)\big]e^{ipx}, \tag{33a}$$

$$j(x,t) = \sum_p\frac{1}{L}\sqrt{K(p)}v(p)\big[\tilde{\rho}_+(p,t) - \tilde{\rho}_-(p,t)\big]e^{ipx}. \tag{33b}$$

Note that $\rho(x,t)$ and $j(x,t)$ can be conveniently expressed using $q_{\pm}^J(p,t)$ and $j_{\pm}^J(p,t)$ in (18).

## 4.1 Euler-scale GHD results

Given the initial conditions $\boldsymbol{\beta}(x,0) = \boldsymbol{\beta}(x)$ defining the state given by $G_{\boldsymbol{\beta}(\cdot)}$ in (19), we found that the thermodynamic fields $\beta_{r',p'}(x,t)$ and $\mu_{r'}^J(x,t)$ in the Euler-scale GHD description in (23) are given by (30). By straightforward computations using (31) and (33), one can show that the corresponding results for heat and charge transport are as follows; a superscripted $\infty$ is used to denote results in the thermodynamic limit, e.g., $\langle\mathcal{O}\rangle_{\boldsymbol{\beta}(x,t)}^{\infty} = \lim_{L\to\infty}\langle\mathcal{O}\rangle_{\boldsymbol{\beta}(x,t)}$.

For heat transport:

$$\langle\mathcal{E}\rangle_{\boldsymbol{\beta}(x,t)}^{\infty} = \sum_r\frac{K(0)\mu_r^J(x - v_{r,0}^{\text{eff}}t)^2}{4\pi v(0)} + \sum_r\int_0^{\infty}\frac{\mathrm{d}p}{2\pi}\frac{\omega(p)}{e^{\beta_{r,p}(x - v_{r,p}^{\text{eff}}t)\omega(p)} - 1} - \int_0^{\infty}\frac{\mathrm{d}p}{2\pi}[v_F - v(p)]p, \tag{34a}$$

$$\langle\mathcal{J}\rangle_{\boldsymbol{\beta}(x,t)}^{\infty} = \sum_r\frac{rK(0)\mu_r^J(x - v_{r,0}^{\text{eff}}t)^2}{4\pi} + \sum_r\int_0^{\infty}\frac{\mathrm{d}p}{2\pi}v_{r,p}^{\text{eff}}\frac{\omega(p)}{e^{\beta_{r,p}(x - v_{r,p}^{\text{eff}}t)\omega(p)} - 1}. \tag{34b}$$

For charge transport:

$$\langle \rho \rangle^\infty_{\boldsymbol{\beta}(x,t)} = \sum_r \frac{K(0)\mu_r^J(x - v_{r,0}^{\mathrm{eff}}t)}{2\pi v(0)}, \qquad \langle j \rangle^\infty_{\boldsymbol{\beta}(x,t)} = \sum_r \frac{rK(0)\mu_r^J(x - v_{r,0}^{\mathrm{eff}}t)}{2\pi}. \qquad (35)$$

Comparing the results above with the CFT ones in [34], we see that those in (35) are the same as in CFT and universal in the sense that they do not depend on the exact details of the NLL interactions: They only depend on the interaction strengths. The same is true for the first terms in (34). However, the second terms in (34) are clearly different from the corresponding CFT results and non-universal in the sense that they do depend on all details of the interactions, including the spatial dependence.

To see that (35) agrees with (4), we note that $\mu_\pm^J(x)$ [conjugate to $q_\pm^J(x)$] are related to the corresponding $\mu_\pm(x)$ [conjugate to $\rho_\pm(x)$] via

$$\mu_\pm^J(p) = \frac{\mu_+(p) + \mu_-(p)}{2} \pm \frac{\mu_+(p) - \mu_-(p)}{2K(p)} \qquad (36)$$

in momentum space [cf. (9) and (13)]. Inserting this into (35) for the case $\mu_\pm(x) = \mu(x)$ reproduces (4).

### 4.2 Comparison with exact analytical results

The precise statement of the Euler-scale hydrodynamic approximation in (21) is as follows: Consider $\langle \mathcal{O}(\lambda x, \lambda t) \rangle_{\boldsymbol{\beta}(\cdot/\lambda)}$ for $\lambda > 0$, then the claim is that

$$\lim_{\lambda \to \infty} \langle \mathcal{O}(\lambda x, \lambda t) \rangle_{\boldsymbol{\beta}(\cdot/\lambda)} = \langle \mathcal{O} \rangle_{\boldsymbol{\beta}(x,t)}. \qquad (37)$$

A formal proof of this for the NLL model is given in Sec. 5.

Below, we instead compare our GHD results for transport in (34) and (35) with the exact analytical results in [31,32] and show that they agree perfectly at the Euler scale. To this end, for our initial state given by (19), let $\beta_{r',p'}(x) = \beta(x)$ for $(r', p') \in \mathcal{I}$ and $\mu_\pm^J(x) = \mu(x)$ [cf. (36)].

The results for heat transport in [32] were for the special case $\mu(x) = 0$ and were given as formal series expansions in the relative height $\epsilon = \delta\beta/\beta$ of the initial inverse-temperature profile $\beta(x) = \beta[1 + \epsilon W(x)]$ for some smooth function $W(x) = W(x + L)$, with only the zeroth- and first-order terms spelled out. For low temperatures, i.e., $\beta > 0$ large, $\epsilon$ is a natural small parameter. Note that while higher-order terms can also be computed, they become increasingly hard to evaluate for the NLL model.[10] Moreover, the results for charge transport in [31] were for the special case $\beta(x) = 0$ and obtained using gauge transformations and no expansions.

Combining the methods in [31] and [32], exact analytical results for both heat and charge transport can be computed, in principle, to all orders in $\epsilon$, even when both $\beta(x)$ and $\mu(x)$ are non-constant. These results are valid at all time and length scales. However, even to first order in $\epsilon$, they are more complicated than the ones in [31,32]. Thus, for simplicity, we give

---

[10]For the case of point-like interactions in [32], all terms could be computed and resummed to explicit formulas in terms of $\beta(x)$ and derivatives thereof, formally meaning that $\epsilon$ did not have to be small. (In that case, $\epsilon$ could more or less be viewed as an accounting tool.)

the explicit formulas only for the particle density and the charge current:

$$\langle \rho(x,t)\rangle^{\infty}_{\beta(\cdot)} = \sum_r \int_{-\infty}^{\infty} \frac{\mathrm{d}p}{2\pi} \frac{K(p)}{v(p)} \frac{\mu(p)}{2\pi} e^{ip[x-rv(p)t]} \tag{38a}$$

$$- \epsilon \sum_r \int_{-\infty}^{\infty} \frac{\mathrm{d}p}{2\pi} \int_{-\infty}^{\infty} \frac{\mathrm{d}p'}{2\pi} W(p) \frac{A(p-p',p')}{v(p-p')} \frac{\mu(-p')}{4\pi} e^{i(p-p')[x-rv(p-p')t]} + O(\epsilon^2),$$

$$\langle j(x,t)\rangle^{\infty}_{\beta(\cdot)} = \sum_r r \int_{-\infty}^{\infty} \frac{\mathrm{d}p}{2\pi} K(p) \frac{\mu(p)}{2\pi} e^{ip[x-rv(p)t]} \tag{38b}$$

$$- \epsilon \sum_r r \int_{-\infty}^{\infty} \frac{\mathrm{d}p}{2\pi} \int_{-\infty}^{\infty} \frac{\mathrm{d}p'}{2\pi} W(p) A(p-p',p') \frac{\mu(-p')}{4\pi} e^{i(p-p')[x-rv(p-p')t]} + O(\epsilon^2)$$

with

$$A(p_1,p_2) = \frac{v(p_1)K(p_2) - v(p_2)K(p_1)}{v(p_2)}, \tag{39}$$

where $\mu(p) = \int_{-\infty}^{\infty} \mathrm{d}x\, \mu(x) e^{-ipx}$ and $W(p) = \int_{-\infty}^{\infty} \mathrm{d}x\, W(x) e^{-ipx}$. One can verify that the first-order terms disappear at the Euler scale, leaving only

$$\lim_{\lambda \to \infty} \langle \rho(\lambda x, \lambda t)\rangle^{\infty}_{\beta(\cdot/\lambda)} = \lim_{\lambda \to \infty} \int_{-\infty}^{\infty} \frac{\mathrm{d}p}{2\pi} \frac{K(p)\lambda\mu(\lambda p)}{2\pi v(p)} \Big[ e^{ip\lambda[x-v(p)t]} + e^{ip\lambda[x+v(p)t]} \Big] + O(\epsilon^2), \tag{40a}$$

$$\lim_{\lambda \to \infty} \langle j(\lambda x, \lambda t)\rangle^{\infty}_{\beta(\cdot/\lambda)} = \lim_{\lambda \to \infty} \int_{-\infty}^{\infty} \frac{\mathrm{d}p}{2\pi} \frac{K(p)\lambda\mu(\lambda p)}{2\pi} \Big[ e^{ip\lambda[x-v(p)t]} - e^{ip\lambda[x+v(p)t]} \Big] + O(\epsilon^2). \tag{40b}$$

Changing variables from $p$ to $p/\lambda$ and using that $K(p/\lambda) \to K(0)$ and $v(p/\lambda) \to v(0)$ as $\lambda \to \infty$ for short-range interactions, it follows that the above agree perfectly with the GHD results in (35) for $\mu_r^J(x) = \mu(x)$, at least up to $O(\epsilon^2)$ corrections.

One can show that the corresponding Euler-scale results for the energy density and the heat current are:

$$\lim_{\lambda \to \infty} \langle \mathcal{E}(\lambda x, \lambda t)\rangle^{\infty}_{\beta(\cdot/\lambda)} = \sum_r \frac{K(0)\mu(x-v_{r,0}^{\mathrm{eff}}t)^2}{4\pi v(0)} + \int_0^{\infty} \frac{\mathrm{d}p}{2\pi} \frac{2\omega(p)}{e^{\beta\omega(p)}-1} - \int_0^{\infty} \frac{\mathrm{d}p}{2\pi}[v_F - v(p)]p$$

$$- \epsilon \sum_r \int_0^{\infty} \frac{\mathrm{d}p}{2\pi} W(x-v_{r,p}^{\mathrm{eff}}t) \frac{\beta\omega(p)^2 e^{\beta\omega(p)}}{\big[e^{\beta\omega(p)}-1\big]^2} + O(\epsilon^2), \tag{41a}$$

$$\lim_{\lambda \to \infty} \langle \mathcal{J}(\lambda x, \lambda t)\rangle^{\infty}_{\beta(\cdot/\lambda)} = \sum_r \frac{rK(0)\mu(x-v_{r,0}^{\mathrm{eff}}t)^2}{4\pi}$$

$$- \epsilon \sum_r \int_0^{\infty} \frac{\mathrm{d}p}{2\pi} v_{r,p}^{\mathrm{eff}} W(x-v_{r,p}^{\mathrm{eff}}t) \frac{\beta\omega(p)^2 e^{\beta\omega(p)}}{\big[e^{\beta\omega(p)}-1\big]^2} + O(\epsilon^2). \tag{41b}$$

(The same formulas but with $\mu(x) = 0$ are obtained from the results in [32] at the Euler scale.) The results above clearly agree with the GHD ones in (34) for $\beta_{r,p}(x) = \beta(x) = \beta[1 + \epsilon W(x)]$ and $\mu_r^J(x) = \mu(x)$, again at least up to $O(\epsilon^2)$ corrections.

Comparing (35) with (38), it is clear that the GHD results are not the same as the exact analytical ones at all time and length scales. In particular, the latter allow for dispersive effects while the former do not. Similar conclusions can be drawn for heat transport by comparing (34) with the corresponding exact analytical ones, where the latter have a more complicated structure than in (34) due to nesting of momenta similar to that seen in the first-order terms in (38), cf. [32]. However, it follows from the above that the GHD results emerge from the exact analytical ones at the Euler scale, at least up to second-order corrections in $\epsilon$ for the inverse-temperature dependence, and formally to all orders in $\epsilon$ from Sec. 5 below.

# 5 Formal proof of the emergence of GHD

In this section, we give a formal proof of (37) for the NLL model.

Consider the expectation in (20) given by $G_{\boldsymbol{\beta}(\cdot)}$ in (19) and the dynamics given by $H$ in (10). Using that the NLL model is invariant under time and spatial translations, we can write $\langle \mathcal{O}(x,t) \rangle_{\boldsymbol{\beta}(\cdot)} = \mathrm{Tr}\big[\mathrm{e}^{-G_{\boldsymbol{\beta}(\cdot)}(-x,-t)} \mathcal{O}(0,0)\big]/\mathrm{Tr}\big[\mathrm{e}^{-G_{\boldsymbol{\beta}(\cdot)}(-x,-t)}\big]$ with

$$
G_{\boldsymbol{\beta}(\cdot)}(x,t) = \sum_{(r',p') \in \mathcal{I}_1} \int_{-L/2}^{L/2} \mathrm{d}x'\, \beta_{r',p'}(x') q_{r',p'}(x'+x,t)
$$
$$
+ \sum_{r'=\pm} \int_{-L/2}^{L/2} \mathrm{d}x'\, \beta_{r',0}(x')\big[q_{r',0}(x'+x,t) - \mu_{r'}^J(x') q_{r'}^J(x'+x,t)\big]. \tag{42}
$$

We are interested in $G_{\boldsymbol{\beta}(\cdot/\lambda)}(\lambda x, \lambda t)$ for large $\lambda > 0$, where we stress that we must also rescale $L$ in the integral by $\lambda$ for consistency. For simplicity, we give details only for the terms involving $q_{r',p'}(x'+x,t)$ for $(r',p') \in \mathcal{I}_1$. Passing to momentum space, we formally obtain

$$
\int_{-\lambda L/2}^{\lambda L/2} \mathrm{d}x'\, \beta_{r',p'}(x'/\lambda) q_{r',p'}(x'+\lambda x, \lambda t) = \sum_p \frac{1}{L} \beta_{r',p'}(-p) q_{r',p'}(p/\lambda, \lambda t) \mathrm{e}^{ipx}
$$
$$
= \sum_p \frac{\pi}{L^2} \beta_{r',p'}(-p) \nu(p')\big[\mathord{:}\tilde{\rho}_{r'}(p/\lambda - p')\tilde{\rho}_{r'}(p')\mathord{:}\, \mathrm{e}^{-ir'[\omega(p/\lambda - p') + \omega(p')]\lambda t}
$$
$$
+ \mathord{:}\tilde{\rho}_{r'}(-p')\tilde{\rho}_{r'}(p/\lambda + p')\mathord{:}\, \mathrm{e}^{-ir'[\omega(-p') + \omega(p/\lambda + p')]\lambda t}\big]\mathrm{e}^{ipx}, \tag{43}
$$

where $\beta_{r',p'}(p) = \int_{-L/2}^{L/2} \mathrm{d}x\, \beta_{r',p'}(x) \mathrm{e}^{-ipx}$ and we used (15a). Since $\omega(p \mp p') + \omega(\pm p') = p\nu^g(p') + O(p^2)$, it follows that

$$
\int_{-\lambda L/2}^{\lambda L/2} \mathrm{d}x'\, \beta_{r',p'}(x'/\lambda) q_{r',p'}(x'+\lambda x, \lambda t) = \sum_p \frac{\pi}{L^2} \beta_{r',p'}(-p) \nu(p')
$$
$$
\times \big[\mathord{:}\tilde{\rho}_{r'}(p/\lambda - p')\tilde{\rho}_{r'}(p')\mathord{:} + \mathord{:}\tilde{\rho}_{r'}(-p')\tilde{\rho}_{r'}(p/\lambda + p')\mathord{:}\big]\mathrm{e}^{ipx - ir'[p\nu^g(p') + O(\lambda^{-1})]t}. \tag{44}
$$

The above formally tends to

$$
\sum_p \frac{2\pi}{L^2} \beta_{r',p'}(-p) \nu(p') \mathord{:}\tilde{\rho}_{r'}(-p')\tilde{\rho}_{r'}(p')\mathord{:}\, \mathrm{e}^{ip[x - r'\nu^g(p')t]} = \beta_{r',p'}(-x + \nu_{r',p'}^{\mathrm{eff}}t) Q_{r',p'} \tag{45}
$$

as $\lambda \to \infty$ with $\nu_{r',p'}^{\mathrm{eff}}$ in (28) and $Q_{r',p'}$ in (12). The corresponding results for the remaining terms are obtained similarly. Combining all of the above formally implies (37) with the r.h.s. given by (22) using $G_{\boldsymbol{\beta}(x,t)}$ in (23) and $\boldsymbol{\beta}(x,t)$ given by (30).

We stress that (43) and the step from (44) to (45) need justification since the operators are not defined for momenta that are not integer multiples of $2\pi/L$ and since the limit is taken on the operator level. One way would be to consider $\exp\big(-G_{\boldsymbol{\beta}(\cdot/\lambda)}(-\lambda x, -\lambda t)\big)$ and $\exp\big(-G_{\boldsymbol{\beta}(x,t)}\big)$ as quadratic forms on a dense subset of our Hilbert space constructed from finite linear combinations of the eigenstates of the Hamiltonian, cf., e.g., [33]. These quadratic forms would then depend on the interactions due to (9) and thus on their short-rangeness. It would be interesting to make the above proof rigorous in this way (or some other) and, in so doing, understand the precise mathematical requirements on the range of the interactions.

# 6 Concluding remarks

In this paper we studied the non-local Luttinger (NLL) model using the recent proposal of Euler-scale generalized hydrodynamics (GHD). Based on the exact solution of this model by

bosonization, we did this as follows: (i) The relevant conserved charges forming our generalized Gibbs ensemble were identified. (ii) The effective velocities (equivalently the flux Jacobians) appearing in the corresponding Euler-scale hydrodynamic equations were derived. (iii) These hydrodynamic equations were solved exactly and shown to have fully explicit solutions that depend non-trivially on the NLL interactions. For completeness, we note that (i) and (ii) are input from the perspective of Euler-scale GHD [3].

The solutions in (iii) were used to compute explicit GHD results for heat and charge transport when evolving in time from initial states defined by inverse-temperature and chemical-potential profiles. Compared with the exact analytical results in [31, 32], which are valid at all time and length scales, the GHD results were observed to be different at small scales but analytically shown to agree perfectly at the Euler scale for short-range interactions. As such, the NLL model can be seen as a simple yet non-trivial tractable example in $1 + 1$ dimensions to analytically study the emergence of hydrodynamics in a quantum many-body system. A general but formal proof of this for the NLL model was given in Sec. 5.

In this paper, we assumed that the interactions were short range. It would be interesting to better understand what happens for long-range interactions, where an emergent Euler-scale hydrodynamic description is not expected on general grounds. [Recall that long-range interactions are allowed by the conditions in (5).] To gain some insight, consider the following example: As mentioned, one possible set of potentials in momentum space are $V_{2,4}(p) = \pi v_F/[1 + (ap)^2]$ with interaction range $a > 0$, and both $v(p)$ and $K(p)$ then also have this momentum dependence [see (7)]. Since $V_{2,4}(p/\lambda)$ tend to 0 as $a \to \infty$ (for $\lambda$ fixed) and $\pi v_F$ as $\lambda \to \infty$ (for $a$ fixed) in the sense of distributions,[11] there is a non-trivial interplay between the Euler scale and sending $a \to \infty$ [cf. (40)]. Indeed, in general, the scaled results depend on the ratio $a/\lambda$, which implies that these limits do not commute. Heuristically, viewing $a \to \infty$ as a long-range limiting case, this suggests that there is no emergent Euler-scale GHD description for the NLL model with long-range interactions.

A proper example of long-range interactions is the unscreened Coulomb potential $V_{2,4}(x) = (\pi v_F/2)(x^2 + x_0^2)^{-1/2}$ with an ultraviolet regularization $x_0 > 0$ studied in [36]. In the infinite volume, this yields $V_{2,4}(p) = \pi v_F K_0(x_0 p)$ in momentum space, where $K_0(\cdot)$ is a modified Bessel function of the second kind, which we recall is singular in $p = 0$. It follows that $V_{2,4}(p/\lambda)$ depend on $x_0 p/\lambda$ and thus diverge as $\lambda \to \infty$, necessitating a more careful analysis.

From the discussion above, the NLL model can also be seen as a simple tractable example to analytically study effects of long-range interactions on the emergence of hydrodynamics.

It would be interesting to investigate the observed differences between the exact analytical results and the GHD ones at smaller time and length scales. One approach would be to include higher-derivative terms in the hydrodynamic approximation [cf. (21)], thereby going beyond Euler-scale hydrodynamics, cf., [10, 11]. (See also Footnote 1 on page 4.) Another interesting problem is to study the hydrodynamic description of the NLL model for the time evolution following an interaction quench, cf., e.g., [31, 37, 38].

As a final remark, we reiterate that one way to make the formal proof in Sec. 5 rigorous is to show that $\exp\bigl(-G_{\beta(\cdot/\lambda)}(-\lambda x, -\lambda t)\bigr)$ tends to $\exp\bigl(-G_{\beta(x,t)}\bigr)$ as $\lambda \to \infty$ in the sense of quadratic forms. We hope to return to this elsewhere. In particular, it would be interesting to understand the precise mathematical requirements on the interactions, including consequences of breaking translation invariance (since our formal proof relies on this), cf. [14].

---

[11]Alternatively, note that, formally, $V_{2,4}(p/\lambda)$ tend to $\pi v_F \delta_{p,0}$ as $a \to \infty$ (for $\lambda$ fixed), which we recall is 0 almost everywhere.

## Acknowledgements:

I am grateful to Vieri Mastropietro for encouraging me to undertake this work and Benjamin Doyon for valuable comments on the manuscript. I also want to thank Denis Bernard, Gian Michele Graf, Gaultier Lambert, and Spyros Sotiriadis for interesting discussions and useful remarks. Financial support from the Wenner-Gren Foundations (No. WGF2019-0061) is gratefully acknowledged.

## A    Solution of the Euler-scale hydrodynamic equations

In this appendix, we show that the solutions to (29) are given by (30).

For $(r', p') \in \mathcal{I}_1$, using that $Q_{r',p'} = \omega(p') n_{r',p'}$ where $n_{r,p} = \tilde{b}_{rp}^+ \tilde{b}_{rp}^-$ is a boson number operator [cf. the discussion preceding (11)] together with the explicit construction of the Hilbert space, see, e.g., [31], we obtain

$$
\langle q_{r',p'} \rangle_{\boldsymbol{\beta}(x,t)} = \frac{1}{L} \langle Q_{r',p'} \rangle_{\boldsymbol{\beta}(x,t)} = \frac{1}{L} \frac{\mathrm{Tr}\big[e^{-\beta_{r',p'}(x,t)Q_{r',p'}} Q_{r',p'}\big]}{\mathrm{Tr}\big[e^{-\beta_{r',p'}(x,t)Q_{r',p'}}\big]}
$$
$$
= \frac{-1}{L} \frac{\partial}{\partial \beta_{r',p'}(x,t)} \log\Big(\mathrm{Tr}\big[e^{-\beta_{r',p'}(x,t)Q_{r',p'}}\big]\Big) = \frac{1}{L} \frac{\omega(p')}{e^{\beta_{r',p'}(x,t)\omega(p')} - 1}. \qquad (46)
$$

Here, we used that $\mathrm{Tr}\big[e^{-\beta_{r',p'}(x,t)Q_{r',p'}}\big] = \sum_{n=0}^{\infty} e^{-n\beta_{r',p'}(x,t)\omega(p')}$. For $p' = 0$, we instead obtain

$$
\langle q_{r',0} \rangle_{\boldsymbol{\beta}(x,t)} = \frac{1}{L} \langle Q_{r',0} \rangle_{\boldsymbol{\beta}(x,t)} = \frac{\pi\nu(0)}{L^2} \frac{\mathrm{Tr}\big[e^{-\beta_{r',0}(x,t)\pi\nu(0)\tilde{Q}_{r'}^2/L} \tilde{Q}_{r'}^2\big]}{\mathrm{Tr}\big[e^{-\beta_{r',0}(x,t)\pi\nu(0)\tilde{Q}_{r'}^2/L}\big]} + \frac{K(0)\mu_{r'}^J(x,t)^2}{4\pi\nu(0)}, \quad (47a)
$$

$$
\langle q_{r'}^J \rangle_{\boldsymbol{\beta}(x,t)} = \frac{1}{L} \langle Q_{r'}^J \rangle_{\boldsymbol{\beta}(x,t)} = \frac{1}{L} \frac{\mathrm{Tr}\big[e^{-\beta_{r',0}(x,t)\pi\nu(0)\tilde{Q}_{r'}^2/L} Q_{r'}^J\big]}{\mathrm{Tr}\big[e^{-\beta_{r',0}(x,t)\pi\nu(0)\tilde{Q}_{r'}^2/L}\big]} + \frac{K(0)\mu_{r'}^J(x,t)}{2\pi\nu(0)}. \qquad (47b)
$$

The first term in (47b) can be shown to be identically zero, while the first term in (47a) can be expressed explicitly with the help of formulas for the Jacobi theta function $\vartheta_3$, see, e.g., Eqs. 16.27.3 and 16.29.3 in [39], with the same prefactor $L^{-2}$ as above. Using this and inserting (46) and (47) into (29) yields

$$
\partial_t \beta_{r',p'}(x,t) + v_{r',p'}^{\mathrm{eff}} \partial_x \beta_{r',p'}(x,t) = 0, \quad \partial_t \mu_{r'}^J(x,t) + v_{r',0}^{\mathrm{eff}} \partial_x \mu_{r'}^J(x,t) = 0. \qquad (48)
$$

Solving these differential equations with the initial conditions $\boldsymbol{\beta}(x,0) = \boldsymbol{\beta}(x)$ yields (30).

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
