# Peer review of "Emergence of generalized hydrodynamics in the non-local Luttinger model"

_SciPost Physics, doi:SciPost Phys. 9, 037 (2020)_

## Round 2 · Referee Report · Anonymous (Referee 1) · 2020-6-12

Strengths

  • the calculations are performed carefully, clearly and almost rigorously, with the non-rigorous steps properly pointed out.
  • It brings out the physics of dispersion and its relation to the Euler scale very well.

Weaknesses

  • The techniques are is restricted to free models

Report

In this paper, the author studies the non-local Luttinger liquid model from the perspective of its non-equilibrium dynamics and connection with generalised hydrodynamics.

The paper is extremely well written. The statement of the problem is clear, the calculations well presented, and the results clear. The problem and results are also interesting; they provide a further understanding of the emergence of hydrodynamics. The problem itself is of physical importance, it is a natural extension of the usual Luttinger liquid model with point-like interaction, and it has received some attention recently especially out of equilibrium. Showing the emergence of hydrodynamics is clearly an interesting problem for this model. The interest, here, is not the effects of interactions - as the model is essentially free (this is so from the viewpoint to the bosons or of the hydrodynamics, but interacting in the original fermions) - but rather the effects of dispersion. It is seen, in quite some detail, how and why dispersions effects, that are present because of the nonzero range of the fermion interaction potentials, go away in the Euler scaling limit. This is expected on physical grounds, but this paper provide quite explicit explanations and results.

A weak point of this paper is that it is restricted to free models. A strong point is that the calculations are performed carefully, clearly and almost rigorously, with the non-rigorous steps properly pointed out. Another strong point is that it brings out the physics of dispersion and its relation to the Euler scale very well.

I found only two typos: page 9 top [(2.6)ff], page 13 top (4.10)ff. Otherwise I do not have additional comment - all concepts are well addressed by the paper.

Requested changes

no change except typos noted in report

  • validity: top
  • significance: good
  • originality: ok
  • clarity: top
  • formatting: perfect
  • grammar: perfect

Author:  Per Moosavi  on 2020-07-17  [id 891]

(in reply to Report 1 on 2020-06-12)

I am thankful to the referee for the report and for the comments.

---

## Round 2 · Referee Report · Anonymous (Referee 2) · 2020-6-20

Strengths

1- Well-written paper, easy to read 2- A clear example of the emergence of Euler scale GHD

Weaknesses

1- Fairly narrow in scope. 2-Contrary to interacting models like hard rods or reversible cellular automata where the emergence of GHD can be understood analytically, the methods used here rely on the non-interacting nature of the model. (of course, the underlying fermionic model is interacting, but at the end of the day, both the GHD and the ab-initio calculations are performed on the bosonized version) 3-The key features of GHD are interactions and the density dependence of the effective velocity, which are unfortunately gone here.

Report

In this paper, the author studies the emergence of generalized hydrodynamics in the non-local Luttinger liquid model. As the non-local Luttinger liquid model is effectively non-interacting, non-equilibrium properties can be computed exactly at all times and length scales. This provides a useful playground to investigate the emergence of Euler scale hydrodynamics. This paper is well-written, and easy to follow. The results are interesting, though as I noted above some of the key ingredients of GHD due to interactions are absent in this model. Still, it's nice to see the emergence of Euler scale GHD, and the discussion of dispersion is valuable.
  • validity: top
  • significance: ok
  • originality: ok
  • clarity: high
  • formatting: excellent
  • grammar: perfect

Author:  Per Moosavi  on 2020-07-17  [id 892]

(in reply to Report 2 on 2020-06-20)

I thank the referee for the report and for the comments.

---

## Round 3 · Author Response

Resubmission with minor adjustments and typos fixed following the referee reports.

---

## Round 3 · List of Changes

1) Typos fixed. 2) Minor stylistic adjustments at a few places. 3) Updated Eqs. (2.8) and (2.11) to make properties manifest; this has no effect on the results or the conclusions. 4) Inessential updates to Eqs. (5.2) and (5.3) to make consistent with the updates described in 3).

---

## Editorial Decision

published